SOFTWARE

# BoReMi: Bokeh-based jupyter-interface for registering spatio-molecular data to related microscopy images

**Jaspreet Ishar[1], Yee Man Tam[2], Simon Mages[2], Johanna Klughammer[2]***

**1** School of Neurobiology, Biochemistry and Biophysics, Tel Aviv University, Tel Aviv, Israel, **2** Gene Center and Department of Biochemistry, Ludwig-Maximilians-Universität, Munich, Germany

\* klughammer@genzentrum.lmu.de

## Abstract

Spatio-molecular data and microscopy images provide complementary information, essential to study structure and function of spatially organised multicellular systems such as healthy or diseased tissues. However, aligning these two types of data can be challenging due to distortions and differences in resolution, orientation, and position. Manual registration is tedious but may be necessary for challenging samples as well as for the generation of ground-truth data sets that enable benchmarking of existing and emerging automated alignment tools. To make the process of manual registration more convenient, efficient, and integrated, we created *BoReMi*, a python-based, Jupyter-integrated, visual tool that offers all the relevant functionalities for aligning and registering spatio-molecular data and associated microscopy images. We showcase *BoReMi's* utility using publicly available data and images and make *BoReMi* as well as an interactive demo available on GitHub.

## Introduction

Following the ground-breaking insights gained through single-cell biology, the necessity to assess the molecular makeup of cells in their native tissue context has been widely recognized. Big consortia such as the Human Tumour Atlas Network propose a dual strategy of single-cell and spatial-profiling [1] to capture biomedically relevant signals, and "spatially resolved transcriptomics" has been pronounced Nature Methods "Method of the year 2020" [2].

A growing number of high-throughput spatio-molecular profiling methods are enabling the biomedical research community to profile tissues at unprecedented molecular and spatial resolution. At the same time, conventional light microscopy in combination with un-targeted tissue stains such as the hematoxylin and eosin (HE) stain remains a primary tool to assess tissue structure in the clinical and research setting for its easy, cheap, and well-established implementation and interpretation. While such microscopy images contain very little molecular information, they are rich in morphologic and phenotypic information and build on more than 100 years of histological research, which in turn is missing for most novel spatio-molecular methods.

The power of combining classical microscopy images and spatio-molecular measurements is evident, but technically only very few spatio-molecular methods (e.g. Spatial Transcriptomics/10x Visium [3]) allow the simultaneous capture of classical microscopy images from

**Data Availability Statement:** Code Availability BoReMi is available on github for download and interactive testing (https://github.com/

jaspreetishar/BoReMi). Data Availability All data and images used in this work have been previously published: Mouse whole brain coronal section Data structured to reproduce the results are available on github: https://github.com/jaspreetishar/BoReMi/blob/main/Binder/sample_spatial_data/mf_obs.csv (cell coordinates and clusters), https://github.com/jaspreetishar/BoReMi/tree/main/Binder/sample_images (images). Mouse hippocampus Slide-seq data: https://singlecell.broadinstitute.org/single_cell/study/SCP815/highly-sensitive-spatial-transcriptomics-at-near-cellular-resolution-with-slide-seqv2#study-download (Puck_200115_08). Human breast cancer metastasis: https://singlecell.broadinstitute.org/single_cell/data/public/SCP2702/htapp-mbc?filename=unregistered_data_bundle.tar.zip (sample 944-7479).

**Funding:** The work was supported by an Else Kröner-Fresenius-Stiftung (EKFS) starting grant (2019_A70) and grants from the German Research Foundation (DFG) (CRC237 and CRC274) awarded to J.K. The funders had no role in study design, data collection and analysis, decision to publish, or preparation of the manuscript.

**Competing interests:** The authors have declared that no competing interests exist.

the same tissue section. Most other popular methods such as MERFISH [4], Slide-seq [5], or CODEX [6] can only rely on the integration with microscopy images from adjacent tissue sections or from publicly available databases such as the Allen Brain Atlas [7].

In order to relate molecular and histological features, a non-trivial registration between the molecular data and the microscopy images is required since they are typically acquired in different scale, orientation, and coordinate space. Elegant computational methods are being developed for the automated alignment of spatio-molecular data. These methods use diverse approaches either modelling transcriptional and physical similarity [8,9], using a landmark-based common coordinate framework [10] or using diffeomorphic metric mapping [11]. While these automated alignment tools each show impressive results and further improvements are to be expected, human interaction and manual alignment will remain necessary for certain cases to improve on automated results, solve ambiguities, or account for higher-level information such as known experimental artefacts. Furthermore, manual alignment will remain important for the generation of ground-truth data that allows to benchmark or even train automated alignment tools. This is similar to how manual segmentation is used as ground truth to train (deep learning) models for automated cell segmentation [12,13]. To date, manual alignments have to be performed using general purpose tools such as ImageJ or Gimp, which necessitates tedious workarounds and decouples the process of manual alignment from the rest of the analysis.

To facilitate and integrate manual alignment of spatio-molecular data and related microscopy images, we created an interactive and user-friendly visual tool, *BoReMi* (**Fig 1**), specifically designed for registering any type of spatio-molecular data to related microscopy images. *BoReMi* is Python-based, enabling all linear (rigid) image and data manipulations (scaling, rotation. translocation, flip) as well as non-linear manipulations, necessary for registration, and provides a graphical user interface within Jupyter notebooks, thus integrating seamlessly with other popular data analysis toolkits.

## Design and implementation

### Design

*BoReMi* is implemented using a powerful Python visualisation library called Bokeh [14] which we extend to conveniently handle spatio-molecular data and microscopy images within Jupyter notebooks. While *BoReMi's* primary functionality is to align one spatio-molecular data section to one microscopy image, it can also simultaneously handle multiple sets of spatio-molecular data of same or different data types, enabling a joint registration. We showcase all functionalities using published data. *BoReMi* can be used online via Binder, on a local computer, and within a compute cluster environment. *BoReMi* is available via GitHub (https://github.com/jaspreetishar/BoReMi).

### Data acquisition

All data has been previously published in the single cell portal (Slide-seq of mouse hippocampus, MERFISH, Slide-seq and HE of a human breast cancer biopsy), the Vizgen Data Resource (MERFISH of mouse brain coronal section), and the Allen Brain Atlas. Detailed links to the data are provided in the data availability statement.

### Framework implementation

To build a framework for the interactive registration of spatio-molecular data and related microscopy images, we used the Python library Bokeh. To implement procedures like parsing

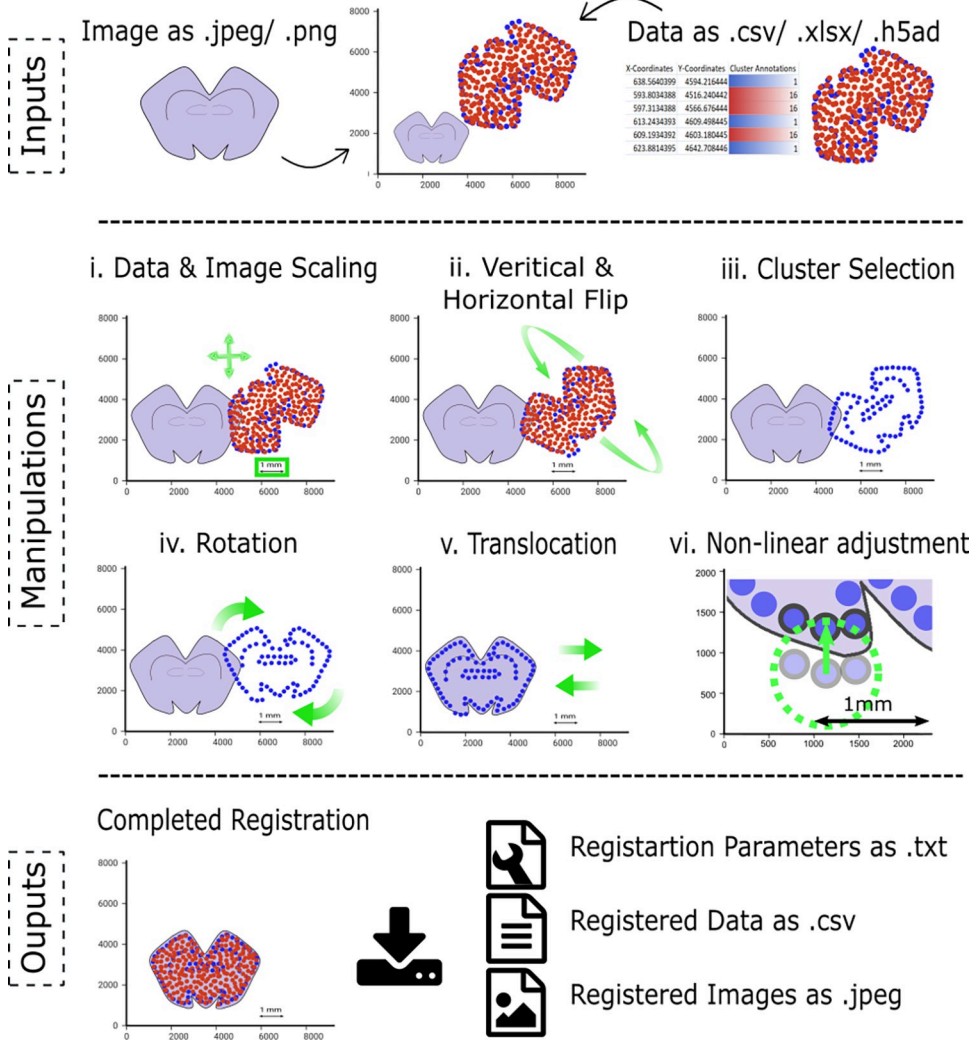

**Fig 1. Schematic illustrating the workflow and showcasing the core functionalities of BoReMi.**

input files and storing the data in a BoReMi-compatible format, the Python libraries AnnData, NumPy and Pandas were used. To enable downloading the registered images, the Python library Pillow was used. BoReMi has modest resource requirements: On a Surface Pro 7 tablet (Intel Core i5-1035G4, 16GB memory) it runs smoothly and opens the MERFISH mouse brain dataset with ~83,000 cells (subsampled to 10,000 cells for display) and images with 100 MP in about two minutes for PNG images and in about 1 minute for JPEG images. Downsampling the number of displayed data points to ~10,000, while all data points are being processed, ensures lag-free interaction even with modest computational resources.

## Downsampling of displayed data points for large datasets

To enhance the performance of BoReMi when aligning a large number of data points, we reduce the number of displayed data points while still maintaining the overall representation of the data. Specifically, the distribution of annotations (e.g., clusters) is maintained using the following formula:

*Variables*:

T = Total number of cells in the original sm-data

$C_i$ = Total number of cells in *i*-th cluster annotation of the original sm-data

$F_i$ = Fraction of the *i*-th annotated cells present in the original sm-data

$D_i$ = Downsampled number of *i*-th annotated cells

N = Total number of cells required in the downsampled data

$$F_i = \frac{C_i}{T}$$

$$D_i = F_i \times N$$

Once the registration process is completed and the updated data is being downloaded, BoR-eMi automatically applies all the manipulations performed by the user to the remaining data points that were not included in the downsampled dataset and thus includes all data points in the updated coordinates file.

If no additional annotation of data points is provided, random downsampling is performed.

*Implementation of data and image manipulations*

Scaling: Data and images are brought to the same scale using the following formula, where "original scale" and "display scale" are input parameters (**Fig 1i**).

For Image:

$$New\ Width = \left( \frac{Original\ Width}{Original\ Scale\ of\ Image} \right) \times Display\ Scale$$

$$New\ Height = \left( \frac{Original\ Height}{Original\ Scale\ of\ Image} \right) \times Display\ Scale$$

For Data:

$$New\ X\ Coordinate = \left( \frac{Original\ X\ Coordinate}{Original\ Scale\ of\ Data} \right) \times Display\ Scale$$

$$New\ Y\ Coordinate = \left( \frac{Original\ Y\ Coordinate}{Original\ Scale\ of\ Data} \right) \times Display\ Scale$$

A scale bar is drawn to represent the display scale as soon as data and image have been resized to the common display scale. The display of the scale-bar is triggered by setting the desired display scale and only if data and image pixel sizes are set correctly, they will be correctly represented by the scale bar.

Horizontal or Vertical flip: To facilitate the mirroring of data about the vertical and the horizontal planes, we employed the rotation angle-dependent formulas given below (**Fig 1ii**). The data undergoes a transformation that includes the reversal of any preceding rotations, if applicable. Subsequently, a mirroring operation is performed about the imaginary axes passing through points ($X_C$, $Y_C$). Following this, the initial rotation is re-applied in the opposite direction. The rotation widget concurrently reflects the rotation in the opposite direction, ensuring seamless synchronisation of flips with any pre-existing rotation:

*Variables*:

$X_O$ = Original X Coordinate

$X_{N'}$ = Temporary X Coordinate

$X_{N''}$ = Final X Coordinate

$X_C$ = X Coordinate of the Central Point of Data Plot

$Y_O$ = Original Y Coordinate

$Y_{N'}$ = Temporary Y Coordinate

$Y_{N''}$ = Final Y Coordinate

$Y_C$ = Y Coordinate of the Central Point of Data Plot

<u>1st Operation:</u>

*Given Rotation Angle = θ*:

$Cos$ = Cos(−θ)

$Sin$ = Sin(−θ)

$$X_{N'} = ((X_O - X_C) \times Cos) - ((Y_O - Y_C) \times Sin) + X_C$$

$$Y_{N'} = ((X_O - X_C) \times Sin) + ((Y_O - Y_C) \times Cos) + Y_C$$

<u>2nd Operation:</u>

Flip about the Vertical Plane:

$$X_{N'} = X_C + (X_C - X_{N'})$$

$$Y_{N'} = Y_{N'}$$

Flip about the Horizontal Plane:

$$X_{N'} = X_{N'}$$

$$Y_{N'} = Y_C + (Y_C - Y_{N'})$$

<u>3rd Operation:</u>

*Given Rotation Angle = θ*:

$Cos$ = Cos(−θ)

$Sin$ = Sin(−θ)

$$X_{N''} = ((X_{N'} - X_C) \times Cos) - ((Y_{N'} - Y_C) \times Sin) + X_C$$

$$Y_{N''} = ((X_{N'} - X_C) \times Sin) + ((Y_{N'} - Y_C) \times Cos) + Y_C$$

<u>Cluster selection</u>: Cluster selection is accomplished by grouping cell coordinates with identical cluster annotations. Initially, all clusters are automatically selected. When a cluster is unselected using the corresponding interactive legend, the corresponding cell coordinates become invisible. Different clusters can be selected individually or collectively, depending on the user's preference. After the completion of the registration process and when the updated coordinates are being downloaded or when a previously unselected cluster becomes re-selected, cells corresponding to unselected cluster annotations reappear, reflecting the same manipulations previously applied to the cells corresponding to selected cluster annotations (**Fig 1iii**).

<u>Rotation</u>: Rotation of the images is achieved by using Bokeh's in-built image manipulation feature. However, to enable rotation of the data, we employed the formulas given below (**Fig 1iv**).

*Variables*:

$X_O$ = Original X Coordinate

$X_N$ = New X Coordinate

$X_C$ = X Coordinate of the Central Point of Data Plot

$Y_O$ = Original Y Coordinate

$Y_N$ = New Y Coordinate

$Y_C$ = Y Coordinate of the Central Point of Data Plot

Input Parameter—Rotation Angle $\theta$:

$Cos = Cos(\theta)$

$Sin = Sin(\theta)$

$$X_N = ((X_O - X_C) \times Cos) - ((Y_O - Y_C) \times Sin) + X_C$$

$$Y_N = ((X_O - X_C) \times Sin) + ((Y_O - Y_C) \times Cos) + Y_C$$

Translocation: To enable translocation of the data in the x and y-axis, we employed the formulas given below (**Fig 1v**):

*Variables*:

$X_O$ = Original X Coordinate

$X_N$ = New X Coordinate

$T_X$ = Translocation value in the x-axis

$Y_O$ = Original Y Coordinate

$Y_N$ = New Y Coordinate

$T_Y$ = Translocation value in the y-axis

$$X_N = X_O + T_X$$

$$Y_N = Y_O + T_Y$$

Non-linear transformations (**Fig 1vi**): Data points can have three states which are switched by a mouse click. Originally, all data points are in a state where they are not selected to be moved. One click enables a data point to be dragged, represented by increased size. Two clicks make a data point hollow and lock it such that it stays in place even if a nearby data point is moved. Three clicks reset to the original state.

Distortion can be applied by dragging an activated (selected) data point. The threshold for distortion, defining the region of influence, is selected by the user (default: 1000 pixels). When a selected data point is dragged, the displacement (dx and dy) from its original position is calculated. The distance of each data point from the selected data point is calculated. If the distance is within the threshold and the data point is not locked, the position of the neighbouring data point is adjusted based on the displacement, scaled by the proximity to the threshold. This ensures that the displacement applied to each neighbouring data point diminishes linearly with the distance from the selected data point, within the specified threshold.

*Variables*:

*index* = Index of the selected data point

*i* = Index of the rest of the data points

$X_{O_{index}}$ = Original X Coordinate of the selected data point $X_{N_{index}}$ = New X Coordinate of the selected data point

$Y_{O_{index}}$ = Original Y Coordinate of the selected data point $Y_{N_{index}}$ = New Y Coordinate of the selected data point

$X_{O_i}$ = Original X Coordinate of the *i*-th data point $X_{N_i}$ = New X Coordinate of the *i*-th data point

$Y_{O_i}$ = Original Y Coordinate of the $i$-th data point $Y_{N_i}$ = New Y Coordinate of the $i$-th data point

$$dx = X_{N_{index}} - X_{O_{index}}$$
$$dy = Y_{N_{index}} - Y_{O_{index}}$$

$D$ = Distance of the $i$-th data point from the selected data point

*Input Parameter*:

$T$ = Threshold for distortion

$$D = \sqrt{\left(X_{O_i} - X_{O_{index}}\right)^2 + \left(Y_{O_i} - Y_{O_{index}}\right)^2}$$

$$X_{N_i} = X_{O_i} + dx \times ((T - D)/T)$$

$$Y_{N_i} = Y_{O_i} + dy \times ((T - D)/T)$$

Undo-Redo functionality (linear transformations): A record log keeps track of all linear manipulations and allows to undo or redo past transformations one by one. This log is saved as a.csv file, when downloading the updated coordinates and the widget parameter file.

Reset functionality (non-linear transformations): This functionality resets all non-linear transformations, restoring the data points to their original positions, retaining only the linear transformations.

*Benchmarking analysis*

STalign v1.0.1 was run following the "Aligning single-cell resolution spatial transcriptomics data to H&E staining image from Visium" tutorial (https://jef.works/STalign/notebooks/merfish-visium-alignment-with-point-annotator.html). The "dx" and "blur" parameters for data rasterization was changed to 10 and 1 respectively. The landmarks on the H&E image and data were annotated manually.

Mouse hippocampus Slide-seq data were filtered to include beads with more than 50 UMIs and genes genes captured by more than 5 beads. Leiden clusters were called using scanpy v1.9.3 (n_neighbors = 5, n_pcs = 40, resolution = 0.3).

For the spatial expression correlation analysis between serial sections, 100x100 μm bins were created and raw counts for each gene were summed up within each bin. Spearman correlation between aggregated counts of matched or randomly paired bins was calculated per gene for each sample pair.

## Exemplary registration of coronal mouse brain sections using linear transformations

Visualising the MERFISH data and HE image together in the view panel revealed their drastic differences in terms of scale, orientation, and position (**Fig 2A, panel i**). To better reveal internal structure, we increased the data point size and selected specific clusters to be displayed (**Fig 2A, panel ii and iii**). To initialise the alignment process, we scaled the data and image by setting the data and image pixel sizes (in μm/pixel), which are properties of the input data and image, as well as the desired display scale to which data and image are scaled and that determines the displayed scale-bar (**Fig 2A, panel iv**). To align the orientation between data and image, we flipped the data vertically using the respective toggle button (**Fig 2A, panel v**). We went on to align the hippocampal structures between image and data by rotating and translocating the data (**Fig 2A, panel vi and viii**). This attempt proved futile since in this view we

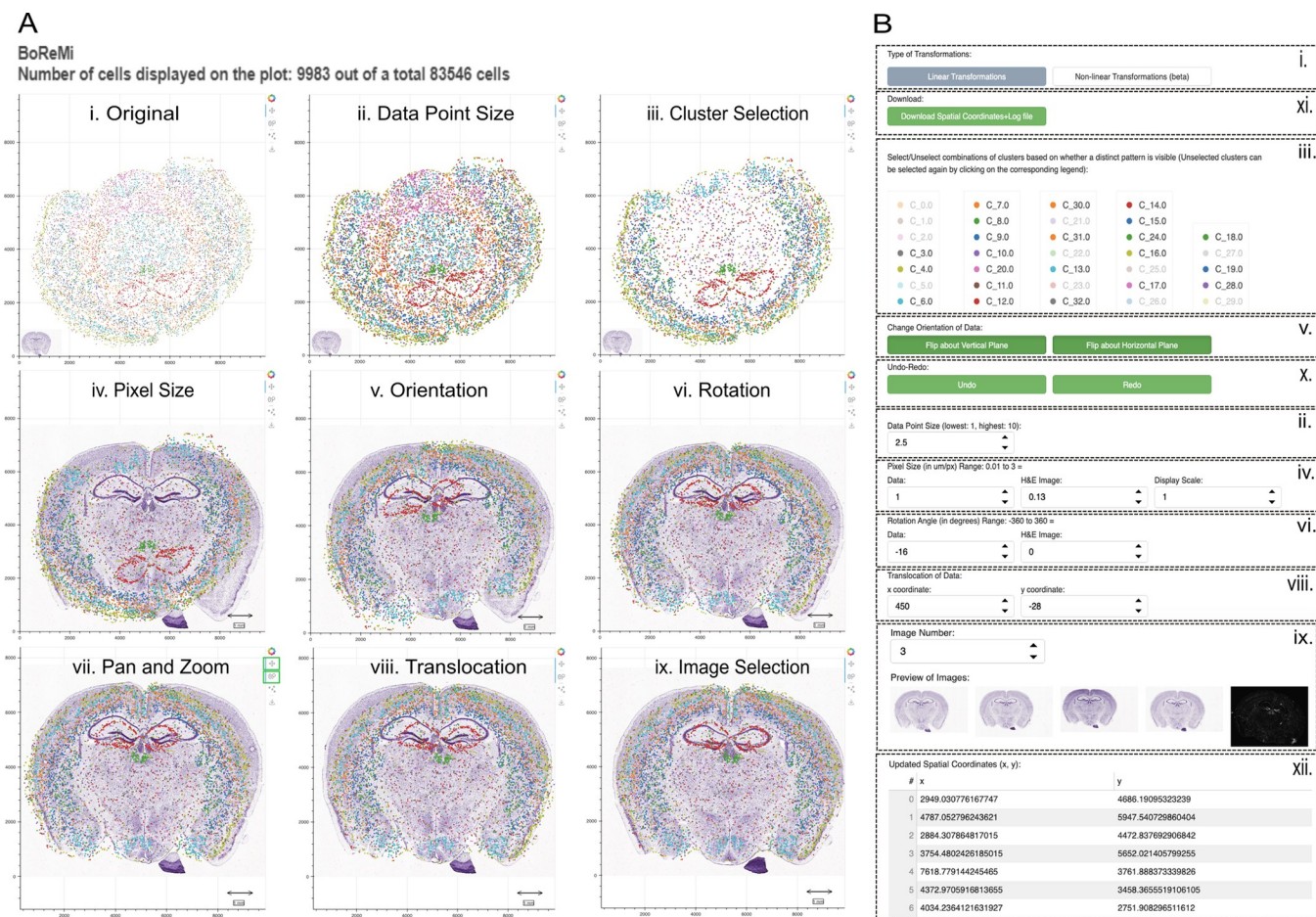

**Fig 2. Exemplary registration of MERFISH data to database HE images using BoReMi's linear transformations. a)** BoReMi's data/image view panel showcasing the different data/image manipulations enabling the registration of MERFISH data and HE images of mouse brain coronal sections. Panel i shows the unmanipulated data as they are obtained from their respective sources. Panels ii-ix show the data after consecutive application of the respective manipulations. **b)** BoReMi's control panel that allows parameter selection for the data/image manipulations as shown in panel a):ii: Data point display size, iii: Data subsetting by cluster selection, iv: Scaling by setting the original resolution for data and image as well as the desired common display scale,v: Data orientation change by flipping, vi: Rotation by setting the angle, vii: Pan and zoom, viii: Translocation by setting x and y changes, and ix: Microscopy image selection. vii (pan and zoom) is controlled through mouse-interaction with the view panel. x shows the undo-redo buttons. xi shows the download buttons for the registered data and image. xii shows the data coordinate live-preview.

could now clearly see that the chosen HE image did not correspond well to the data. Before selecting a new image from *BoReMi's* image gallery (**Fig 2A, panel ix**), we zoomed out a little to get a better overview and a bit more margin (**Fig 2A, panel vii**). The newly selected image fit the data well (considering that HE and data did not come from serial sections), and by small further rotation and translocation adjustments, we completed the registration between spatio-molecular data HE image (**Fig 2A, panel ix**). Finally, *BoReMi* allows users to download the registered data and image together with a registration file that contains all applied transformation parameters for future use or reference (**Fig 2B**). In this example, the HE images already had a good position and orientation, so their angle, position, and orientation did not need to be adjusted. Similarly, the MERFISH data already came at a scale of 1μm per pixel, which was equal to the chosen display scale and thus did not need adjustment. In other scenarios a different set of manipulations might be needed.

## Results

*BoReMi's* workflow and core functionalities are illustrated in **Fig 1**. As inputs for the spatio-molecular data, *BoReMi* supports.csv,.xlsx, and.h5ad file formats containing spatial x-y coordinates of the observations as mandatory information and categorical annotations (e.g. cluster or cell type assignments) as optional information. As inputs for microscopy images BoReMi accepts the.jpeg and.png formats. After the data is loaded, the spatio-molecular data as well as the microscopy images can be manipulated to achieve alignments at a common, user-defined resolution. BoReMi allows all possible linear (rigid) manipulations, including scaling, flipping, rotation, and translocation as well as non-linear (elastic) adjustments of local data points. To reveal certain landmark structures, it is possible to colour and temporarily subset the spatio-molecular data based on categorical annotations. When the alignment is complete, the registered microscopy image and spatio-molecular data are saved to disk in.jpeg and.csv format, respectively, together with a record of the registration parameters in.txt format (**Fig 1**). We illustrate the utility and applicability of *BoReMi* by registering publicly available MERFISH data from a coronal mouse brain section to corresponding HE images from the Allen Brain Atlas achieving good approximate registration based on visual inspection using linear transformations (**Fig 2**). Since data and HE images are not from serial sections but rather the HE is the best fitting available image, a perfect registration is not to be expected. To further improve the alignment by "reattaching" some disconnected data points (possibly caused by a tear), we used BoReMi's non-linear point-wise manipulation feature (**Fig 3A**), which allows to individually place selected data points and their neighbours in an elastic manner. This feature should be used with great caution since it changes the relative position of data points (similar to warping in automated methods) and thus can severely bias downstream analysis. To streamline the registration process, non-linear manipulations can only be applied after linear alignment is complete since they are intended for fine adjustments. Linear manipulations can be used in an arbitrary order and repeatedly, but we recommend prioritising scaling, because it allows operating at a defined scale and the display of a scale-bar.

To demonstrate BoReMi's support of "cell-matching", we aligned the MERFISH data to a DAPI image generated from the same tissue slice (**Fig 3B**). This fine-tuned registration was achieved by zooming in and finely adjusting the linear registration parameters, until MERFISH cells and DAPI signals overlapped.

To demonstrate BoReMi's capability of registering data of multiple sections to the same HE image, we used serially sectioned MERFISH (3 slices) and Slide-Seq (1 slice) data from a metastatic breast cancer biopsy (**Fig 4A**). All slices were individually registered to a common HE image, accommodating individual differences in position and scale. In contrast to the mouse brain section (**Fig 2**), these sections showed very little internal structure but a clear outer contour and they had been cut in a serial manner, theoretically enabling near-perfect registration. We quantified the goodness of the achieved registration by pairwise spatial correlation of gene expression across matched 100x100μm bins (considering spatially variable and in-variable genes) and through the overlap between data and HE. As expected, we found high agreement for the MERFISH samples (median $\rho$ = 0.24–0.31 and 95.2% - 96.3% HE overlap) and lower but still considerable agreement for the Slide-seq sample (mean $\rho$ = 0.04–0.12 and 70.6% HE overlap).

Finally, we registered a published mouse hippocampus Slide-seq sample using a state-of-the-art automated registration tool (STalign) and found that the lack of outer contour as well as internal density structure made it hard to set appropriate landmarks. This ultimately led to global distortion with part of the data not overlapping the HE, while still achieving excellent local registration of the hippocampal structures. BoReMi on the other hand allowed

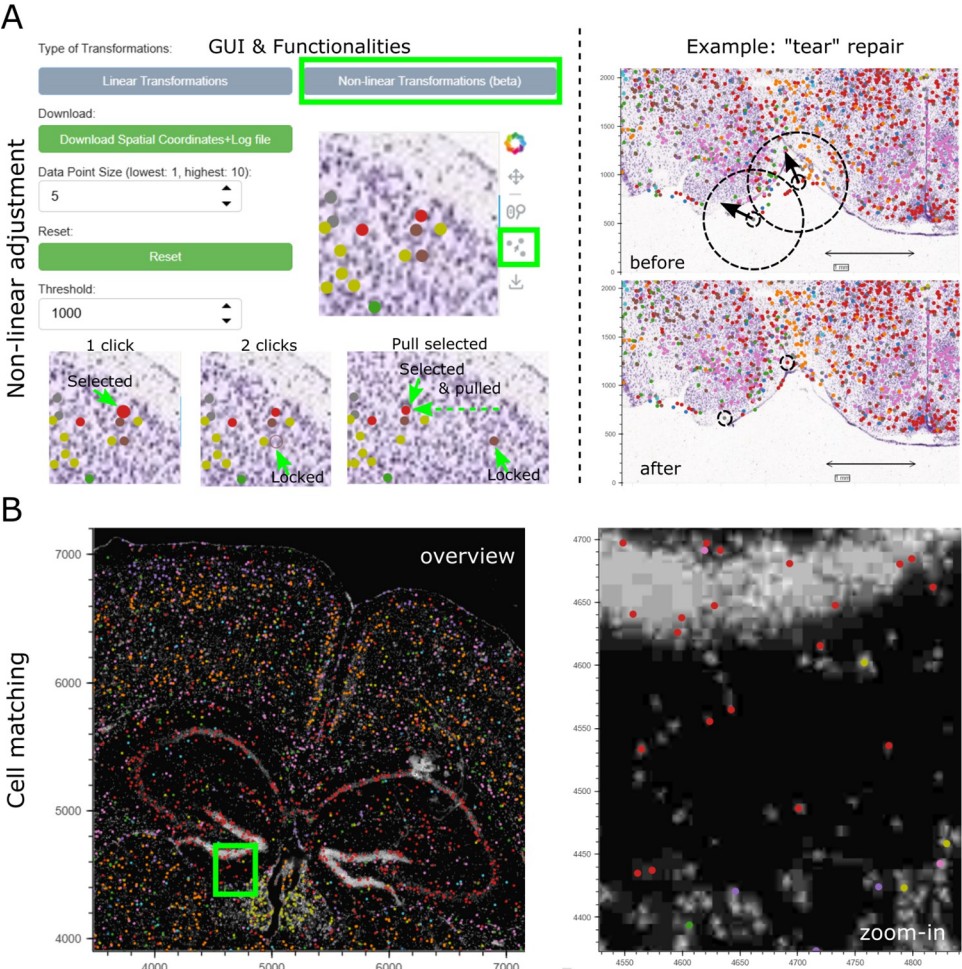

**Fig 3. Fine-tuning of alignments. a)** Left: Non-linear manipulations control panel and functionalities. Manipulations are performed through mouse-interactions after activating the "point-select" widget; Right: Example for adjusting the position of slightly disconnected data points (possibly due to a tear) using BoReMi's elastic non-linear transformation, building on the linear registration in Fig 2. The dragged cells as well as the "threshold of influence" (here set to 500 μm) are indicated by dashed circles. Arrows indicate the drag direction. **b)** Example for highly precise alignment which allows cell matching between cell-segmented data (as in panel a) and images (here DAPI) of the same tissue section by high magnification (zoom-in) and fine positional adjustment.

convenient undistorted registration using cluster-based internal structure and only linear transformations (**Fig 5**).

While manual alignment remains a relatively tedious process especially in comparison to automated approaches, in some cases it is necessary as demonstrated here. *BoReMi* makes this process significantly more convenient, by integrating in the analysis environment and providing all functionalities needed for the designated task, as well as convenient data import and export capabilities.

## Availability and future directions

Here, we present *BoReMi* (https://github.com/jaspreetishar/BoReMi), an interactive, Jupyter-embedded tool for convenient manual alignment and registration of spatio-molecular data and associated microscopy images. We demonstrate *BoReMi's* features and applicability using publicly available data and images featuring coronal mouse brain sections and biopsies of

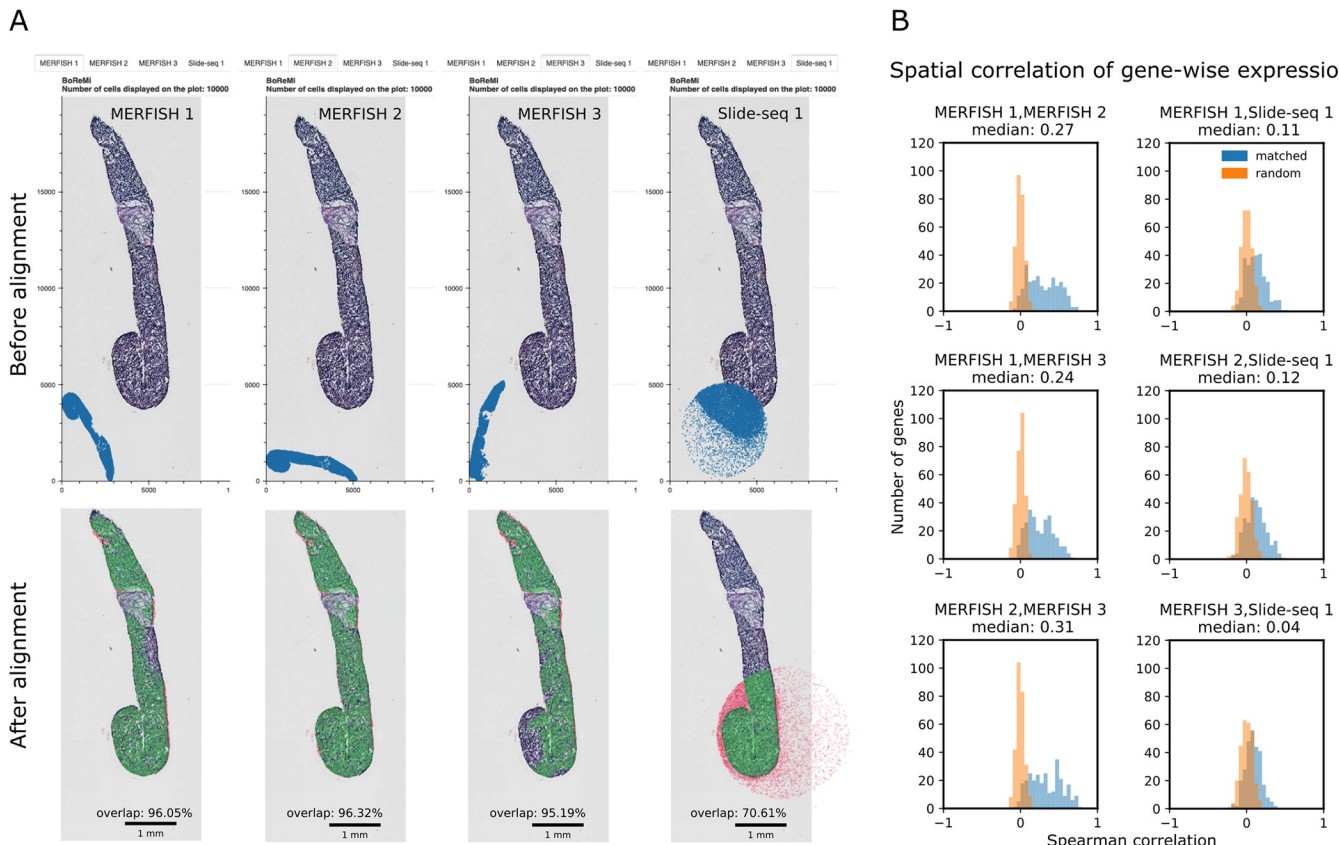

**Fig 4. Registration of multiple serial sections processed with MERFISH or Slide-seq. a)** Top: Data and HE images before alignment as displayed by BoReMi, with different sections accessible through different tabs. Bottom: Data and HE images after alignment. The percent overlap of data and HE is indicated and non-overlapping data points are highlighted in red. **b)** Histogram depicting spatial Spearman correlation of gene expression calculated for each gene across matched or randomly paired 100x100μm bins between pairs of tissue sections (as indicated).

breast cancer metastasis profiled with MERFISH and Slide-seq. Through *BoReMi*, all necessary image and data manipulations including linear and non-linear manipulations can be performed as part of standard python analysis environments within Jupyter, seamlessly integrating the alignment process with the rest of the analysis. These features, together with convenience functionalities such as zoom/pan, data display settings, undo-redo functionality, and flexible input/output options make alignments with *BoReMi* highly efficient. Future improvements include speed-up in data and image loading and transformation, continuous value display of data points, more parameters for non-linear adjustments, data annotation functionalities, and integration with a python-based image viewer such as napari.

While automated data-to-image registration is efficient and convenient, development of such tools requires good ground-truth datasets for training and benchmarking. To date, majorly indirect measures of alignment success, such as region-specific cell type compositions or spatial similarity in expression between serial sections, are being used. In analogy to the field of (cell) segmentation where manual segmentation has traditionally served as ground-truth, manual registration may serve as direct ground-truth for data-to-image registration. *BoReMi* makes the creation of comprehensive expert-aligned ground-truth datasets less daunting and, as such, sets the foundation for appropriate benchmarking and comparison of automated registration tools. Finally, *BoReMi* provides an efficient option for cases in which automated registration performs suboptimally calling for manual adjustments or when

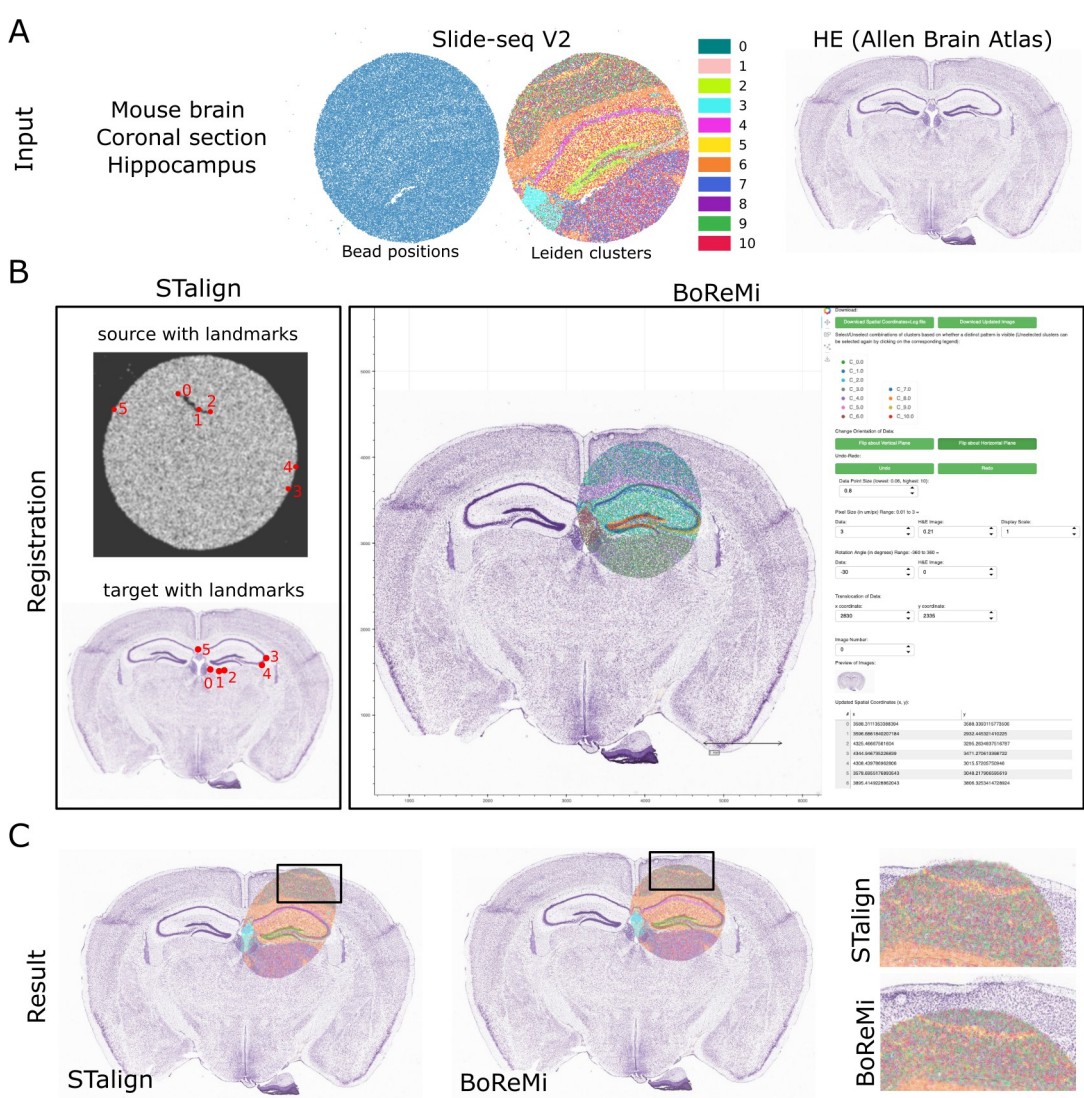

**Fig 5. Alignment of mouse hippocampus Slide-seq data to HE with STalign and BoReMi. a)** Overview of the inputs Slide-seq puck and an Allen Brain Atlas HE image. Leiden clusters (resolution 0.3) are indicated and show internal structure, while bead distribution (beads > 50 UMIs) itself does not show structure. **b)** STalign and BoReMi visual interface and setup. STalign landmarks were set to the best of our ability. **c)** Final alignment results produced by STalign and BoReMi, respectively. The region of over-distortion by STalign is amplified for both alignments.

automated registration cannot be used due to experimental setups that obscure structure or inherently structure poor tissues.

## Acknowledgments

We gratefully acknowledge LMU Klinikum for providing computing resources on their Clinical Open Research Engine (CORE) and the Bioinformatic Core Facility of the Biomedical Center Munich for providing computing resources on their HPC system.

## Author Contributions

**Conceptualization:** Johanna Klughammer.

**Data curation:** Jaspreet Ishar, Johanna Klughammer.

**Formal analysis:** Jaspreet Ishar, Yee Man Tam, Simon Mages.

**Funding acquisition:** Johanna Klughammer.

**Investigation:** Jaspreet Ishar, Johanna Klughammer.

**Methodology:** Jaspreet Ishar, Yee Man Tam, Simon Mages, Johanna Klughammer.

**Project administration:** Johanna Klughammer.

**Resources:** Jaspreet Ishar, Johanna Klughammer.

**Software:** Jaspreet Ishar.

**Supervision:** Johanna Klughammer.

**Validation:** Yee Man Tam, Simon Mages.

**Visualization:** Jaspreet Ishar, Yee Man Tam, Johanna Klughammer.

**Writing – original draft:** Jaspreet Ishar, Johanna Klughammer.

**Writing – review & editing:** Jaspreet Ishar, Yee Man Tam, Simon Mages, Johanna Klughammer.

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
