## [Decision Letter · Decision Letter 0]

3 Apr 2024

Dear Prof .Dr. Klughammer,

Thank you very much for submitting your manuscript "BoReMi: Bokeh-based jupyter-interface for Registering spatio-molecular data to related Microscopy images" for consideration at PLOS Computational Biology.

As with all papers reviewed by the journal, your manuscript was reviewed by members of the editorial board and by several independent reviewers. In light of the reviews (below this email), we would like to invite the resubmission of a significantly-revised version that takes into account the reviewers' comments.

We cannot make any decision about publication until we have seen the revised manuscript and your response to the reviewers' comments. Your revised manuscript is also likely to be sent to reviewers for further evaluation.

Sincerely,

Lewis M. Antill

Guest Editor

PLOS Computational Biology

Ilya Ioshikhes

Section Editor

PLOS Computational Biology

Reviewer's Responses to Questions

**Comments to the Authors:**

Reviewer #1: Ishar et al. have developed an innovative python-based, interactive visual tool, BoReMi, for the manual alignment of spatial molecular data to images through rigid transformations. This tool emerges as a particularly valuable resource amid the growing adoption of spatial molecular profiling technologies, offering an integrative and interactive solution that stands out in the current landscape of similar software. BoReMi's functionality is both timely and highly beneficial, manuscript written clearly and code is clear. I recommend its publication after minor edits.

My comments below, largely reflective of enhancements for a potential version 2, are intended to further distinguish BoReMi from competing methodologies. Additionally, I suggest several improvements to the manuscript for a more impactful presentation of BoReMi's capabilities:

1- Non-linear Transformations: The tool currently supports rigid transformations, showcasing exceptional performance with large imaging files. This feature is invaluable for initial alignment tasks. However, for comprehensive utility, it would be beneficial if the discussion section could recommend non-linear alignment tools that complement BoReMi for subsequent alignment steps.

2- Streamlining the Results Section: The middle part of the Results section, which details a specific analysis process, could be condensed or moved to supplementary materials or an online guide. This adjustment would allow the main text to more effectively highlight BoReMi's versatile features and their applicability across various scenarios.

3- Visualization Capabilities: It would be valuable if the authors could discuss whether BoReMi can display cluster assignments in different colors and how it utilizes the inherent molecular information of spatial transcriptomics data. Additionally, the ability to handle and visually represent continuous molecular expression data would significantly enhance BoReMi. These points can be added as how to or in discussion.

4- Computational Performance: The manuscript should detail the computational requirements of BoReMi, especially how it scales with the size of the molecular data and image database. Given the substantial size of Merscope files, ensuring that user has the right tools to run BoReMi.

Minor Points:

- The manuscript hints at the potential for processing multiple spatio-molecular data sets simultaneously but does not delve into this feature. Clarification on the concurrent visualization and independent manipulation of different data sets would be beneficial.

- Introducing a record log of all performed transformations would aid reproducibility and enable users to replicate alignments with other tools.

- Further explanation on whether users can select which data set serves as the reference during alignment and the possibility of applying transformations in reverse would enhance BoReMi's flexibility.

In its current form, we will likely use BoReMi as the first step to check alignment pairs before applying elastic like tools. These suggestions aim to refine both BoReMi's functionality and its presentation which are already at a very high standard.

Reviewer #2: The manuscript from Ishar, Tam and Klughammer describes the tool BoReMi for aligning spatial molecular data (i.e., spatial distributions of transcripts or proteins) to imaging data such as from Haematoxylin & Eosin stains of tissue. Such alignment is key if one is to make biological sense of molecular data, since these data typically lack information about tissue structure. Further, more and more such spatio-molecular datasets are likely to be generated and analysed in the next years. Tools in this sphere are therefore important and timely, and as such this paper is potentially of wide interest. BoReMi is easy to use, doesn’t need coding experience, and has a nice interactive demo – these are all strengths.

In its present form however, the manuscript is very minimalist. To us it reads as more of an announcement of the tool than as a research paper. There are several points we think should be addressed before the paper can be published in PLoS Computational Biology.

Major points

1. Overall, the authors should make a better case for this tool. As they state, there are existing tools for doing such alignments. For instance, STAlign was recently published (PMID: 38065970) and landmark-based affine transformation has been used for some time. The authors cite existing approaches, but they don’t compare the performance of their tool to the state-of-the-art, or discuss in any depth the unique use cases for BoReMi-based alignment beyond the need for generating manual ground truth. At a minimum, this should be argued in more detail. Even better, the authors should compare performance to existing approaches and possibly even show an instance of how BoReMi alignment can be used to solve a problem or address a question inaccessible with existing approaches.

2. A single example of data integration with a single type of molecular data (MERFISH data) is given. This example is furthermore not at single-cell resolution and uses very structured image data (i.e., brain tissue, with alignment based on hippocampus structure). Does BoReMi work for single cells? For tissue with more homogenous structure (e.g., tumors)? Does it work for data types other than MERFISH? The manuscript and potential users would benefit from some more examples and a clearer assessment of cases where this approach is/is not likely to be beneficial.

3. Importantly, there is no assessment of how well the manual alignment with BoReMi works, neither any sort of sanity check (e.g., do well studied cell types or tissue regions show the expected molecular profiles?), nor any quantitative assessment. The authors are proposing that alignments generated by BoReMi can be used as ground truth to quantitatively check automated alignment methods; isn’t it crucial then that the manual alignment is accurate? How would the authors assess this in their own work and how would a user of BoReMi be expected to do so?

4. The manuscript states (lines 228-229) that cell type assignments or clusters are optional, but this did not seem to be the case when we tested the tool. These annotations appear mandatory, meaning that preprocessing of the molecular data is necessary to use BoReMi. What are the requirements for this preprocessing? There is no information provided on input file format specifications for the molecular data (e.g., csv with cell ids, center cell coordinates, other?); any requirements should be described in detail. If this type of preprocessing or annotation is not mandatory, the authors should then show how the alignment works without it.

5. How does BoReMi deal with any need for non-linear transformations, or combinations of linear and non-linear transformations? How does it deal with things like tissue tears or folds?

6. From a user perspective, it would be extremely practical to have a reset/undo button in BoReMi, for instance if one makes a mistake or decides to try a different strategy during alignment. At present it seems one must re-do the full alignment process in such cases.

7. In the README on GitHub, inexperienced users would benefit from a more detailed explanation of how to create the conda-environment (e.g. conda create -n) and how to install respective package from the .txt (e.g. conda install --file requirements.txt). In general, the more tutorial-like information that can be generated for users, the better.

Minor points

8. On lines 250-252, the manuscript refers to a new image having to be selected. This is confusing. Isn’t it the case that there is a single image (i.e., the consecutive section) that needs to be aligned to the molecular (MERFISH) data? Why should one be able to select one out of many images?

9. It was not possible to reproduce the results from the paper using the settings shown in Figure 2. Perhaps these settings were not intended for this purpose, but we suggest that the values shown in the figure should then be corrected, or the correct settings given somewhere clearly in the paper (if the tool is indeed deterministic).

10. Cluster selection no longer works once regions are selected with the toolbar (e.g. box selection) – is this a bug in the software?

11. The step sizes used for adjusting the data point size and the x/y coordinates are very small. Is there a reason for this? Is it tailored for single-cell matching? That would make it all the more compelling to show such an example.

12. It may be useful to include a small example dataset directly in the GitHub repository.

13. The authors could consider integrating this with popular python-based image viewers such as napari (as a plugin).

14. It would be helpful to mention minimal computational requirements for smooth running of BoReMi if any (e.g., memory use). Also, we also found no information on whether the software would be frozen or continually updated; it would be good to know in regard to software unit testing and validation checks.

**Have the authors made all data and (if applicable) computational code underlying the findings in their manuscript fully available?**

Reviewer #1: Yes

Reviewer #2: Yes

PLOS authors have the option to publish the peer review history of their article (what does this mean?). If published, this will include your full peer review and any attached files.

Reviewer #1: **Yes: **Ozgun Gokce

Reviewer #2: **Yes: **Dr. Natalie de Souza
---

## [Decision Letter · Decision Letter 1]

18 Sep 2024

Dear Prof .Dr. Klughammer,

We are pleased to inform you that your manuscript 'BoReMi: Bokeh-based jupyter-interface for Registering spatio-molecular data to related Microscopy images' has been provisionally accepted for publication in PLOS Computational Biology.

Best regards,

Lewis M. Antill

Guest Editor

PLOS Computational Biology

Ilya Ioshikhes

Section Editor

PLOS Computational Biology

We thank the authors for their revisions, to which the reviewers find the manuscript much improved and is now ready for publication after two minor corrections (please see comments from Reviewer 2).

Reviewer's Responses to Questions

**Comments to the Authors:**

Reviewer #1: I thank the authors for their thorough revisions and responses. The BoReMi tool effectively addresses the crucial need for registering spatio-molecular data to microscopy images, showcasing its versatility in this area. The manuscript is clearly written, the code is robust, and the figures are well-executed. I strongly support the publication of this elegant work.

Reviewer #2: I find the manuscript much improved and recommend publication.

Two v. small things:

- in the legend of Figure 3b (which is very nice), I found the term 'cell-segmented' odd since it isn't what is typically referred to as cell segmentation

- in the legend of Figure 4a, I would make clear that the molecular data is blue in the 'before alignment' panel and green in the 'after alignment' panel.

**Have the authors made all data and (if applicable) computational code underlying the findings in their manuscript fully available?**

Reviewer #1: Yes

Reviewer #2: Yes

PLOS authors have the option to publish the peer review history of their article (what does this mean?). If published, this will include your full peer review and any attached files.

Reviewer #1: **Yes: **Ozgun Gokce

Reviewer #2: **Yes: **Natalie de Souza

---

## [Editor Report · Acceptance letter]

30 Sep 2024

PCOMPBIOL-D-23-02043R1 

BoReMi: Bokeh-based jupyter-interface for Registering spatio-molecular data to related Microscopy images

Dear Dr Klughammer,

I am pleased to inform you that your manuscript has been formally accepted for publication in PLOS Computational Biology. Your manuscript is now with our production department and you will be notified of the publication date in due course.

With kind regards,

Anita Estes
